# Development of a novel complex inflammatory bowel disease mouse model: Reproducing human inflammatory bowel disease etiologies in mice

Sun-Min Seo[ID][1], Na-Won Kim[1], Eun-Seon Yoo[1], Ji-Hun Lee[1], Ah-Reum Kang[1], Han-Bi Jeong[1], Won-Yong Shim[1], Dong-Hyun Kim[1], Young-Jun Park[1], Kieun Bae[2], Kyong-Ah Yoon[2], Yang-Kyu Choi[ID][1]*

1 Department of Laboratory Animal Medicine, College of Veterinary Medicine, Konkuk University, Seoul, Republic of Korea, 2 Department of Veterinary Biochemistry, College of Veterinary Medicine, Konkuk University, Seoul, Republic of Korea

* yangkyc@konkuk.ac.kr

## Abstract

Inflammatory bowel disease (IBD), caused by environmental factors associated with the host's genetic traits, is represented by Crohn's disease and ulcerative colitis. Despite the increasing number of patients with IBD, the current treatment is limited to symptomatic therapy. A complex IBD model mimicking the human IBD etiology is required to overcome this limitation. Herein, we developed novel complex IBD models using interleukin 2 receptor subunit gamma (Il2rg)-deficient mice, high-fat diet, dextran sodium sulfate, and *Citrobacter rodentium*. The more IBD factors applied complexly, colon length shortened and inflammation worsened. The levels of pro-inflammatory cytokines increased with increased IBD factors. Anti-inflammatory cytokine decreased in all factors application but increased in Il2rg deficiency and Westernized diet combination. Additionally, the pro-inflammatory transcription factors and leaky intestinal epithelial marker were upregulated by a combination of IBD factors. Species diversity decreased with IBD factors. Phylogenetic diversity decreased as IBD factors were applied but increased with combined Il2rg deficiency and Westernized diet. The more IBD factors applied complexly, the more severe the dysbiosis. Finally, we developed a novel complex IBD model using various IBD factors. This model more closely mimic human IBD based on colonic inflammation and dysbiosis than the previous models. Based on these results, our novel complex IBD model could be a valuable tool for further IBD research.

## 1. Introduction

Inflammatory bowel disease (IBD), represented by Crohn's disease (CD) and ulcerative colitis (UC) is a chronic inflammatory disease of the gastrointestinal tract. CD and UC are caused by environmental factors, such as a Westernized diet [1–3], changes in the intestinal flora [4, 5],

**Data Availability Statement:** All relevant data are within the manuscript.

**Funding:** This work was supported by the Bio & Medical Technology Development Program of the National Research Foundation (NRF) funded by the Korea government (MSIT) (Nos. 2020R1A2C2005898 and 2021M3H9A1097269) and the Konkuk University Researcher Fund in 2023. the funders had no role in study design, data collection and analysis, decision to publish, or preparation of the manuscript.

**Competing interests:** The authors have declared that no competing interests exist.

and chemical stress [6, 7], depending on the host's genetic traits [4, 5]. The prevalence of IBD has increased annually due to industrialization and overnutrition. Therefore, the incidence rate, previously high in the United States, Europe, and Oceania, shows an accelerating prevalence rate in Asia, South America, and Africa [8].

Despite the increasing number of patients with IBD and the social burden due to the westernization of emerging countries, only symptomatic treatment using 5-aminosalicylic acid, corticosteroids, immunosuppressants, anti-tumor necrosis factor-α (TNF-α) agents, and bowel resection has been used as treatment methods for IBD. However, immunosuppressants, anti-TNF-α agents, and bowel resection are burdensome for lifelong patients because of their side effects and invasive nature [9]. Since fecal microbiota transplantation (FMT) was first used to treat *Clostridium difficile* infection in humans, it has been studied as an alternative therapeutic strategy [10]. However, FMT is still limited to clinical trials for IBD treatment [11].

Research on fundamental IBD therapeutics and their etiology is ongoing to overcome these limitations, and these studies require animal models that mimic the pathogenesis of human IBD. Currently developed animal models that mimic human IBD etiology include diet-induced [12, 13], bacteria-induced [14–16], chemical-induced [7, 14, 17–19], and genetically engineered models [20, 21].

However, many IBD models so far have limitations such that they do not mimic the complex pathogenicity of IBD in humans but only fragmentarily mimic the human IBD etiology. We hypothesized that combining various IBD factors could create an IBD model that mimics the current complex pathogenicity of human IBD. Therefore, we used high-fat diet to mimic Westernized diet [2, 3]; *Citrobacter rodentium*, a human model of EHEC and EPEC, to mimic pathogenic bacterial infection [14]; Dextran sodium sulfate (DSS) to mimic chemical stress [2, 14]; and interleukin 2 receptor subunit gamma (Il2rg)-deficient mice, also known as the common gamma chain, to mimic genetic traits [22]. We tested a combination of these IBD factors in a mouse model to test this hypothesis.

## 2. Methods

### 2.1. Experimental design

Eight-week-old wild-type C57BL/6J mice and Il2rg-deficient mice, generated using the CRISPR/Cas9 genome editing system in a previous study [22], were divided into C57BL/6J wild-type mice (WT) (n = 12), Il2rg-deficient mice fed with normal-diet (ND) (n = 12), Il2rg-deficient mice fed with high-fat diet (HFD) (n = 12), and Il2rg-deficient mice treated with high-fat diet, *C. rodentium*, and DSS (HCD) (n = 13). Furthermore, mice were fed with a 60% kcal rodent diet (D12492, Research Diets Inc., New Brunswick, NJ, USA) or a normal rodent diet (Teklad 2018S, Inotiv, West Lafayette, IN, USA) *ad libitum* for 56 days (8 weeks). After the application of each diet, the WT, ND, and HFD groups were monitored for an additional 17 days. In the HCD group, after 56 days of 60% kcal rodent diet administration, $1.6 \times 10^8$ CFU of *C. rodentium* (ATCC 51459, ATCC, Manassas, VA, USA) was suspended in Luria-Bertani broth and administered with single intragastric inoculation and 1% DSS (Dextran sodium sulfate salt, colitis grade, MP Biomedicals, Irvine, CA, USA) for 7 days, *ad libitum*. After administration, the HCD group was monitored for an additional 10 days (Fig 1). All groups of mice were observed daily. Mice were immediately euthanized if they experienced 20% body weight loss. One mouse in the ND group was euthanized due to reaching the humane endpoint criteria by malocclusion. The euthanized mouse in the ND group was excluded from the analysis. No animals died before meeting humane endpoint criteria. Environmental enrichment including wood chew block and pulp house (Woojung Bio, Republic

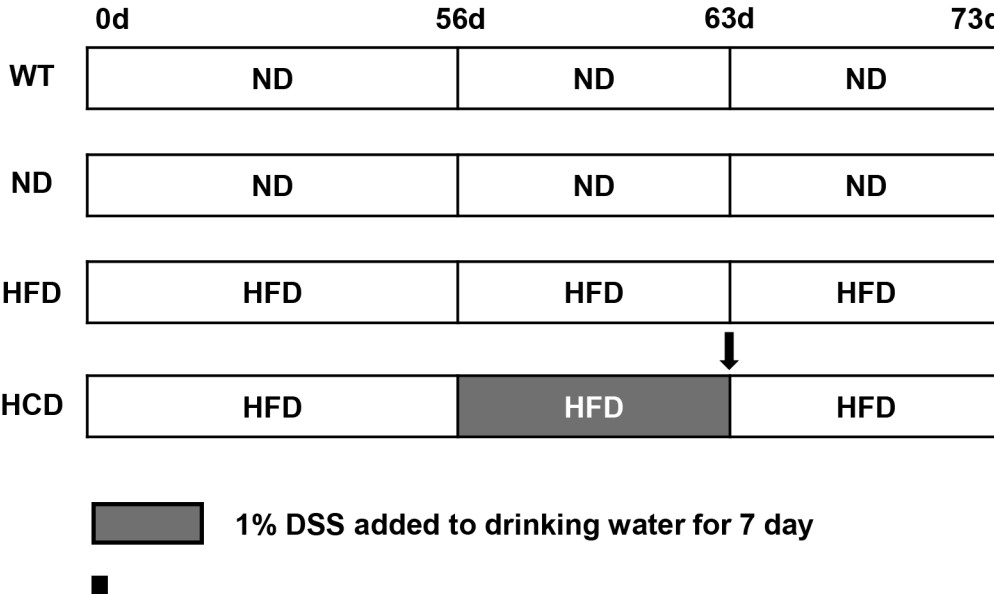

**Fig 1. Schematic diagram of the experimental design of a complex IBD mouse model.** Eight-week-old wild-type C57BL/6J and Il2rg-deficient mice were divided into C57BL/6J wild-type (WT) group (n = 12), Il2rg-deficient mice fed with normal diet (ND) group (n = 11), Il2rg-deficient mice fed with high-fat diet (HFD) group (n = 12), and Il2rg-deficient mice treated with high-fat diet, *C. rodentium*, and DSS (HCD) group (n = 13). Furthermore, a 60% kcal rodent diet or a normal rodent diet was fed to each group for 56 days (8 weeks). In the HCD group, $1.6 \times 10^8$ CFU of *C. rodentium* was administered using single intragastric inoculation, and 1% of DSS was administered for 7 days, ad libitum. After administration, the HCD group was monitored for an additional 10 days. The WT, ND, and HFD groups were monitored for an additional 17 days after 56 days of diet application.

of Korea) were provided to diminish distress. All procedures were performed by trained researchers and in accordance with the Konkuk University Institutional Animal Care and Use Committee (IACUC) and ARRIVE guidelines and regulations. The study was approved by Konkuk University IACUC (KU22048).

## 2.2. Histopathology

Whole colons were harvested from mice after euthanasia. Euthanasia was performed using $CO_2$ with a flow rate of 30% to 70% of chamber volume/min in compliance with AVMA guidelines. Each group of colons was trimmed using the Swiss roll technique, followed by fixation with 10% neutral-buffered formalin. After the routine processing and paraffin-embedded procedure, the tissues were cut into 4 um sections and stained with hematoxylin and eosin (H&E). The sections were evaluated using a BX53 light microscope (Olympus, Tokyo, Japan), and images were captured using a DP74 system (Olympus). The histopathological severity of each group of colons was scored using criteria modified from a previous study [23]. The modified histopathological criteria are presented in Table 1. The ratio of the inflamed area to the total colonic area in each group was also calculated. Histopathological analyses were performed independently on three separate researchers.

## 2.3. Myeloperoxidase assay

The supernatant obtained by homogenizing the colon tissue was measured for myeloperoxidase (MPO) using the myeloperoxidase activity assay kit (Colorimetric) (Ab105136, Abcam,

**Table 1. Histopathological scoring criteria for IBD.**

| Parameter | Score | Criteria |
|---|---|---|
| **Severity of inflammation** | 0 | Rare inflammatory cells in the lamina propria |
| | 1 | Increased numbers of granulocytes in the lamina propria |
| | 2 | Confluence of inflammatory cells extending into the submucosa |
| | 3 | Transmural extension of the inflammatory infiltrate |
| **Crypt damage** | 0 | Intact crypts |
| | 1 | Loss of the basal one-third |
| | 2 | Loss of the basal two-thirds |
| | 3 | Entire crypt loss |
| **Erosion** | 0 | Absence of erosion |
| | 1 | 1 to 3 foci of erosion |
| | 2 | 4 to 6 foci of erosion |
| | 3 | Confluent or extensive erosion |
| **Ulceration** | 0 | Absence of ulceration |
| | 1 | 1 to 3 foci of ulceration |
| | 2 | 4 to 6 foci of ulceration |
| | 3 | Confluent or extensive ulceration |

Cambridge, UK) according to the manufacturer's instructions. The results are shown as units per milligram of the weight of tissue.

## 2.4. Cytokine evaluation

The colon tissue was homogenized using FastPrep-24 5G (MP Biomedicals). After homogenization, total RNA was extracted using the *MagListo*™ 5M Universal RNA Extraction Kit (Bioneer, Daejeon, Republic of Korea) and reverse-transcribed to cDNA using Maxime™ RT PreMix (Oligo dT Primer, iNtRon Biotechnology, Gyeonggi, Republic of Korea). Quantitative real-time polymerase chain reaction (qPCR) was performed using AccuPower® 2X GreenStar™ qPCR Master Mix (Bioneer). The cycling procedure was as follows: pre-denaturation at 95 ºC for 5 min, denaturation at 95 ºC for 15 s, and annealing/extension at 60 ºC for 30 s. Forty cycles were performed using a CFX96 Touch Real-Time PCR Detection System (Bio-rad, Hercules, CA, USA). Target cytokines and specific primers used are listed in Table 2. To detect relative changes in mRNA levels of interferon gamma (IFN-γ), interleukin-1 beta (IL-1β), interleukin-6 (IL-6), interleukin-10 (IL-10), C-X-C motif chemokine ligand 10 (CXCL10), and tumor necrosis factor- α (TNF-α) cytokine mRNA levels of each sample were normalized using glyceraldehyde 3-phosphate dehydrogenase (GAPDH).

## 2.5. Enzyme-linked immunosorbent assay (ELISA)

Enzyme-linked immunosorbent assay (ELISA) was performed to measure the levels of inflammatory and anti-inflammatory cytokines in each group of mice. The cytokines including interferon-gamma (IFN-γ), interleukin-1 beta (IL-1β), interleukin-6 (IL-6), interleukin-10 (IL-10), C-X-C motif chemokine ligand 10 (CXCL10), and tumor necrosis factor-α (TNF-α) were quantified using the Duoset ELISA kit (R&D systems, Minneapolis, MN, USA) following the manufacturer's instructions. Serum from mice was collected and tested without dilution. Absorbance was measured using the SpectraMax microplate reader (Molecular Devices, San Jose, CA, USA).

**Table 2. Specific primers for the target cytokines.**

| 1 | GAPDH | F | 5'- AAC TTT GGC ATT GTG GAA GG -3' |
|---|---|---|---|
| | | R | 5'- ACA CAT TGG GGG TAG GAA CA -3' |
| 2 | IFN-γ | F | 5'- TCA AGT GGC ATA GAT GTG GAA GAA -3' |
| | | R | 5'- TGG CTC TGC AGG ATT TTC ATG -3' |
| 3 | Il-1β | F | 5'- TCG CTC AGG GTC ACA AGA AA -3' |
| | | R | 5'- CAT CAG AGG CAA GGA GGA AAA C -3' |
| 4 | Il-6 | F | 5'- ACA AGT CGG AGG CTT AAT TAC ACA T -3' |
| | | R | 5'- TTG CCA TTG CAC AAC TCT TTT C -3' |
| 5 | Il-10 | F | 5'- ACC TGG TAG AAG TGA TGC CCC AGG CA -3' |
| | | R | 5'- CTA TGC AGT TGA TGA AGA TGT CAA A -3' |
| 6 | CXCL10 | F | 5'- TTG TGC GAA AAG AAG TGC AG -3' |
| | | R | 5'- TAC AAA CAC AGC CTC CCA CA -3' |
| 7 | TNF-α | F | 5'- AGG CTG CCC CGA CTA CGT -3' |
| | | R | 5'- GAC TTT CTC CTG GTA TGA GAT AGC AAA -3' |

## 2.6. Western blot

Proteins were extracted from the colon (5-10mg) using RIPA buffer (1X, Cell Signaling Technology, Beverly, MA, USA) supplemented with phosphatase inhibitor cocktail (1X, Cell signaling Technology), protease inhibitor (1X, Quartett, Berlin, Germany), dithiothreitol (2 mM, Thermo Scientific, Waltham, MA, USA), and phenylmethanesulfonyl fluoride (1 mM, Sigma-Aldrich, Brulington, MA, USA). Protein concentration was calculated using a Pierce™ BCA Protein Assay Kit (Thermo Scientific). Equal amounts of protein (10 ug per lane) were separated by SDS-polyacrylamide gel electrophoresis and transferred to Immobilon-P nitrocellulose membranes (Merck, Kenilworth, NJ, USA). The membranes were blocked with 5% skim milk in TBS with 1% Trixon X-100 (TBST) for 1 hour at room temperature and then incubated overnight at 4°C with the primary antibodies against β-actin (A1978, 1:5000; Sigma-Aldrich), phopho-NF-κB p65 (MAB-15160, 1:1000; Thermo Scientific), and Claudin-2 (32–5600, 1:1000; Life Technologies, Carlsbad, CA, USA). Proteins were visualized using a horseradish peroxidase-conjugated secondary antibody and enhanced chemiluminescence reagent (Bio-Rad Laboratories). The intensities of the bands were visualized by using ImageQuant LAS 4000 (GE Healthcare, Little Chalfont, England) and Image Lab software (V 6.0.1, Bio-rad).

## 2.7. Extraction and sequencing of DNA

The samples used for microbiome analysis, the outer mucus of the colon, were stored at -80 ºC in a deep freezing condition. The samples were sent to LAS (Gimpo, Republic of Korea) for total DNA extraction, next-generation sequencing (NGS) library preparation, and sequencing. Dropsense96 (Trinean, Gent, Belgium) was used for quality control and accurate DNA quantification of each sample. After DNA quantification, NGS library preparation for the V3-V4 region was performed using the 16S Metagenomic Sequencing Library preparation (Illumina, San Diego, CA, USA). Quality control for the NGS library was conducted using Fragment Analyzer (Agilent Technologies, Santa Clara, CA, USA) and dsDNA 910 Reagent Kit, 35 bp–1,500 bp (Agilent Technologies). Sequencing was subsequently conducted using the MiSeq Sequencing System (Illumina) with 300 bp paired-end reads.

## 2.8. Biome analysis

Demultiplexed sequences were filtered by sequence quality, and a feature table was generated after chimeric reads were removed using the DADA2 plugin [24]. Naïve Bayes classifiers pre-trained with reference database (Greengenes2 2022.10) [25, 26] were used to assign taxonomy to each sequence. Full analysis, including the determination of alpha and beta analyses, was performed using the QIIME2 program [27], and comparative analyses were performed using LEfSe [28]. Data visualization was conducted using Dokdo (1.16.0) [29].

## 2.9. Statistical analysis

Data are presented as mean ± standard error of the mean (SEM). Comparison between 4 groups (WT, ND, HFD, and HCD group) was performed using the Kruskal-Wallis test. The permutational multivariate analysis of variance (PERMANOVA) was performed for beta diversity analyses. All statistical analyses were performed using GraphPad Prism 9.5.1 (Graph-Pad Software Inc., San Diego, CA, USA). P<0.05 was considered statistically significant.

## 3. Results

### 3.1. Gross lesion and histopathological differences induced by the combination of IBD-affecting factors

Combinations of factors known to affect IBD have been used to reproduce the disease in mice, mimicking human IBD. The genetic factors were imitated using Il2rg-deficient mice generated by the CRISPR/CAS9 genome editing system in our previous study [22]. In addition, a high-fat diet, *C. rodentium*, and DSS were co-administered to mimic a Westernized diet and environmental factors. The C57BL/6J mice with a genetic background of Il2rg-deficient mice were used as a control group for comparison.

After 73 days of the experiment procedure, the colons were harvested, and their lengths were measured (Fig 2). Gross lesions and colon shortening were not observed in the WT and ND groups. However, macroscopic shortening was observed HFD and HCD groups. Colonic hypertrophy was observed in the HCD group (Fig 2A). The intestines were measured to confirm the difference in colon length between each group. The mean colon length was 8.38 cm, 8.42 cm, 7.35 cm, and 5.38 cm in the WT, ND, HFD, and HCD groups, respectively. There

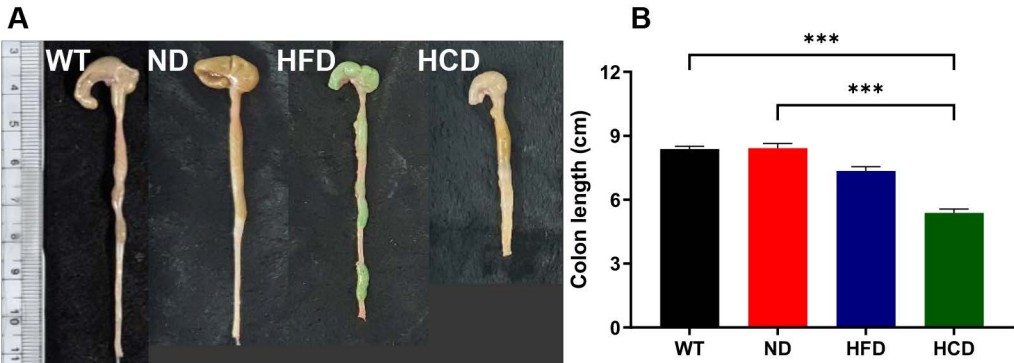

**Fig 2. Colon length changes according to IBD-exacerbated factors in Il2rg-deficient mice.** Independent high-fat diet application or high-fat diet with *C. rodentium* and DSS combined application to Il2rg-deficient mice shortened the colon, compared with the C57BL/6J WT and Il2rg-deficient mice ND groups. (**A**) Representative colon lengths of each group. (**B**) The colon length of the HFD and HCD groups was significantly shortened compared with the WT and ND groups. The data for each group is represented as the mean ± SEM. (n = 11–13) ***P < 0.001. (Kruskal-Wallis test).

was no significant difference in colon length between the WT and ND groups. However, significant differences were observed between the WT and HCD (P<0.001), and ND and HCD (P<0.001) (Fig 2B).

Based on the gross lesion results, histopathological examination and scoring of the colon in each group were performed. Histopathological findings showed that inflammatory cell infiltration was absent in the WT and ND groups, similar to the gross lesions. However, unlike the WT and ND groups, the HFD group showed infiltration of inflammatory cells in the lamina propria. Moreover, in the HCD group, an increase in the number of inflammatory cells in the mucosa and submucosa, transmural inflammation, loss of epithelial crypts, and epithelial ulceration were observed (Fig 3A). Histopathological findings were scored based on the criteria listed in Table 1 to compare the pathological findings in each group. In contrast to the WT and ND groups, which had scores of 0, the mean values of the HFD and HCD groups were 0.67 and 4.50, respectively. The WT and ND groups showed a significant difference from the HCD group (P<0.01 and P<0.001, respectively) (Fig 3B). Also, the ratio of the inflamed area to the total area was calculated. The percentage of inflamed areas also showed similar results to the histopathological scoring in the colon. Those of the WT and ND groups were 0%; however, those of the HFD and HCD groups were 2.23% and 7.79%, respectively. Significant differences were observed between the WT and HCD groups and the ND and HCD groups (P<0.01) (Fig 3C). In addition, the colonic MPO activity, a specific biomarker for IBD, was remarkably higher in the HCD group, depending on the severity of inflammation. MPO activity of the WT, ND, and HFD groups were 0.75, 0.97, and 2.21 unit/mg, respectively. However, in the HCD group, MPO activity was 6.08 unit/mg. Significant differences were observed between the WT and HCD groups (P<0.001) and the ND and HCD groups (P<0.01) (Fig 3D).

## 3.2. Inflammatory and anti-inflammatory cytokine expression differed depending on the combination of IBD-affecting factors

Based on the gross lesions and histopathological differences between each group, pro-inflammatory and anti-inflammatory cytokines were analyzed to determine how these factors affect inflammation in the colon. mRNA expression levels of pro-inflammatory cytokines, including interferon-gamma (IFN-γ), interleukin-1 beta (Il-1β), interleukin 6 (Il-6), C-X-C motif chemokine ligand 10 (CXCL10), TNF-α, and mRNA expression level of anti-inflammatory cytokine interleukin 10 (Il-10) were analyzed from colon tissue.

Even though relative levels of IFN-γ increased as IBD occurred, no significant differences were observed between each group (Fig 4A). Relative levels of Il-1β also increased, depending on the IBD factor. The HCD group showed 1.44, 1.31, and 1.14 times higher Il-1β levels than the WT, ND, and HFD groups respectively, but significant difference was observed only in WT and HCD group comparison (P<0.01) Fig 4B). The relative levels of Il-6 increased as IBD progressed. The ND group showed 0.66 times lower IL-6 levels than the WT group; however, no significant difference was observed The HCD group showed 2.18, 3.82, and 2.58 times higher Il-6 levels than the WT, ND, and HFD groups respectively, but significant difference was observed only in ND and HCD group comparison (P<0.01) (Fig 4C). The relative levels of CXCL10 also increased as the IBD factor was applied. The HCD group showed 1.24, 1.18, and 1.08 times higher CXCL10 levels than the WT, ND, and HFD groups respectively, but significant difference was observed only in WT and HCD group comparison (P<0.05) (Fig 4E). Relative levels of TNF-α increased, depending on the IBD factor application. The HCD group showed 1.53, 1.26, and 1.17 times higher TNF-α levels than the WT, ND, and HFD groups respectively, but significant difference was observed only in WT and HCD group comparison (P<0.01) (Fig 4F). The relative levels of Il-10, an anti-inflammatory cytokine,

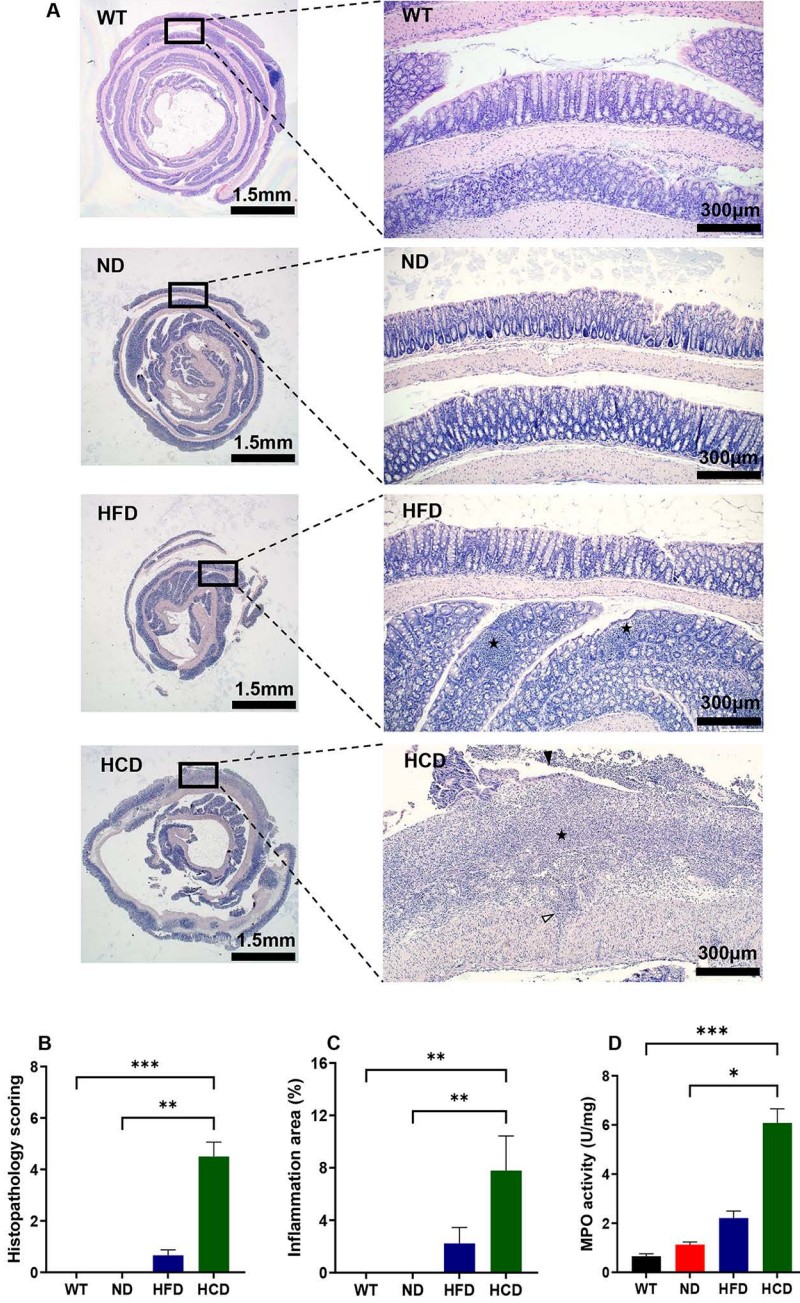

**Fig 3. Histopathological analysis, histopathological scoring, and MPO activity of the colons of each group.** (**A**) Representative sections of H&E stained the colon of each group. Swiss rolls of whole colons (left: magnification x12.5) were magnified to assess levels of colitis (right: magnification x100); Inflammation (black star); transmural inflammation (white arrowhead); ulceration (black arrowhead). (**B**) The colon of each group was scored following the scoring criteria indicated in Table 1. (n = 5–6) (**C**) The ratio of the inflamed area to the total colonic area for each group. (n = 5–6) (**D**) Myeloperoxidase (MPO) activity in colon tissues. (n = 6–7). The data for each group is represented as the mean ± SEM. *P < 0.05; **P < 0.01; ***P < 0.001. (Kruskal-Wallis test).

were considerably higher in the HFD group and lower in the HCD group than those in the WT and ND groups. The WT, ND and HFD groups showed 3.19, 2.99, and 6.28 times higher Il-10 levels than the HCD group and the HFD group showed 1.97 and 2.1 times higher Il-10

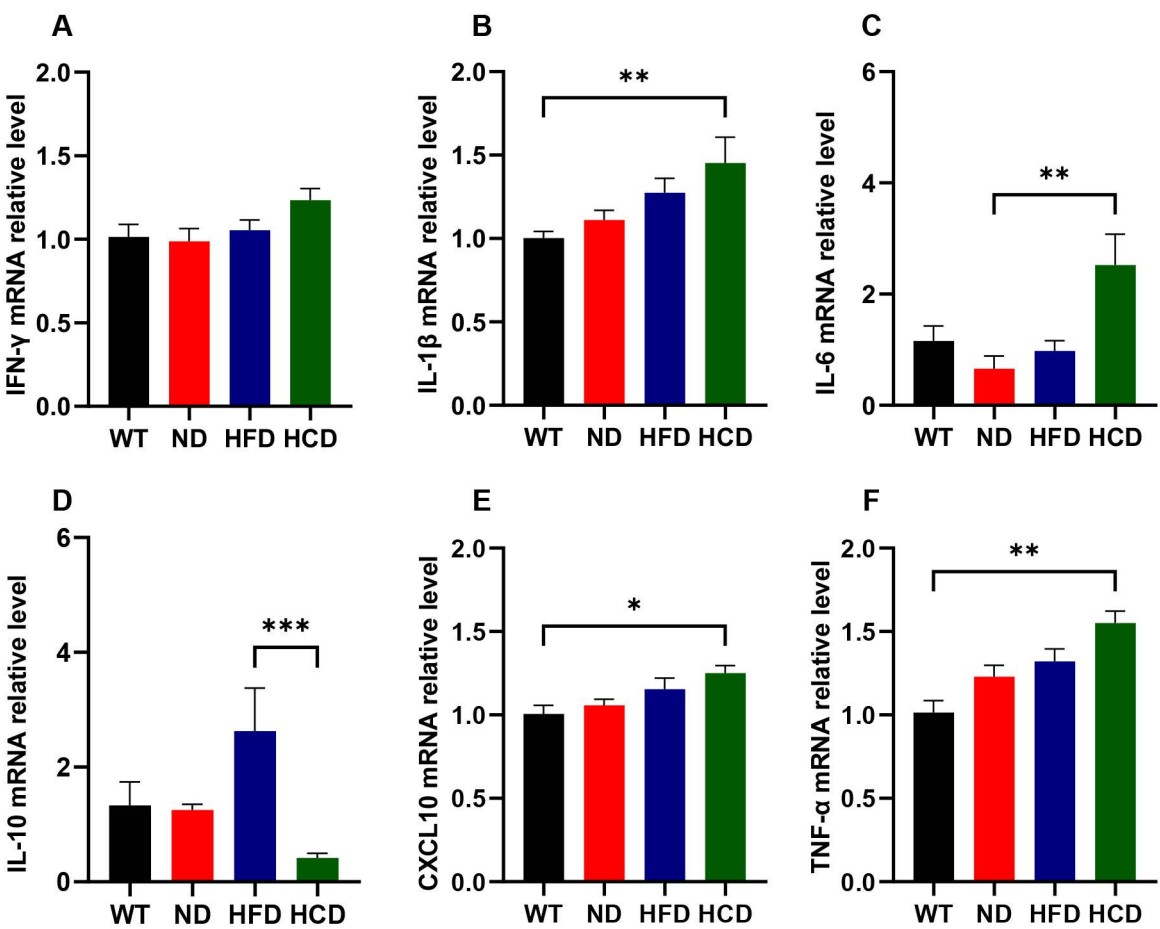

**Fig 4. The relative expression levels of the pro-inflammatory and anti-inflammatory cytokine.** mRNA levels of pro-inflammatory and anti-inflammatory cytokines, including IFN-γ (**A**), Il-1β (**B**), Il-6 (**C**), Il-10 (**D**), CXCL10 (**E**), and TNF-α (**F**), were quantified from the colon tissue of each group of mice. Each mRNA level of cytokines was normalized with GAPDH, the reference gene. The data for each group is represented as the mean ± SEM. (n = 6–7) *P < 0.05; **P < 0.01; ***P < 0.001. (Kruskal-Wallis test).

levels than the WT and ND groups respectively. The significant difference only was observed between HFD and HCD group comparison (P<0.001) (Fig 4D).

To assess the reliability of mRNA expression level results of inflammatory cytokines, the protein expression levels of the cytokines, including IFN-γ, Il-1β, Il-6, CXCL10, TNF-α, and Il-10 were also analyzed. Although IFN-γ did not show a significant difference in mRNA levels, protein levels of IFN-γ showed significant changes as IBD occurred. The HCD group showed 1.38, 1.34, and 1.2 times higher than the WT, ND, and HFD groups. The significant differences were observed in WT and ND groups were compared with the HCD group (P<0.05) (Fig 5A). The protein levels of Il-1β increased. The HCD group showed 1.89, 1.34, and 1.17 times higher Il-1β levels than the WT, ND, and HFD groups respectively, but significant difference was observed only in WT and HCD group comparison (P<0.05) (Fig 5B). The protein levels of Il-6 dramatically increased. The protein levels of Il-6 dramatically increased. The HCD group showed 3.86, 3.88, and 3.06 times higher Il-6 levels than the WT, ND, and HFD groups. The significant differences were observed in WT and ND groups were compared with the HCD group (P<0.01) (Fig 5C). The protein levels of CXCL10 also increased as the IBD factor was applied. The HCD group showed 1.47, 1.35 and 1.22 times higher CXCL10

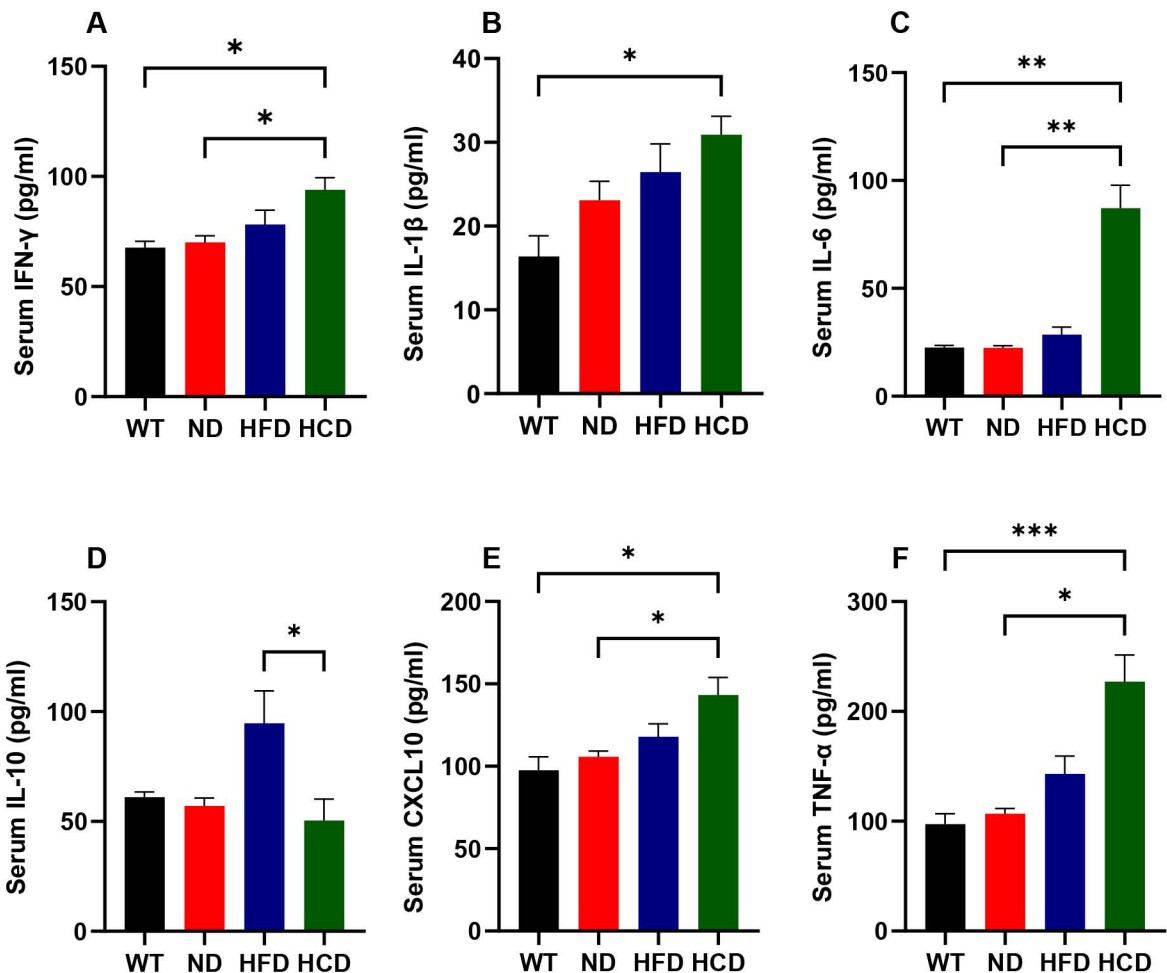

**Fig 5. The protein levels of the pro-inflammatory and anti-inflammatory cytokine.** protein levels of pro-inflammatory and anti-inflammatory cytokines, including IFN-γ (**A**), Il-1β (**B**), Il-6 (**C**), Il-10 (**D**), CXCL10 (**E**), and TNF-α (**F**), were quantified from the serum of each group of mice. Each level of cytokines was measured through ELISA. The data for each group is represented as the mean ± SEM. (n = 6–7) *P < 0.05; **P < 0.01; ***P < 0.001. (Kruskal-Wallis test).

levels than the WT, ND, and HFD groups. The significant differences were observed in WT and ND groups were compared with the HCD group (P<0.05) (Fig 5E). The protein expression of TNF-α increased in the HCD group. The HCD group showed 2.33, 2.13 and 1.59 times higher TNF-α levels than the WT, ND, and HFD groups. The significant differences were observed in WT and ND groups were compared with the HCD group (P<0.001, P<0.01) (Fig 5F). The protein levels of an anti-inflammatory cytokine, Il-10 showed the same trend as mRNA expression. The WT, ND and HFD groups showed 1.21, 1.13, and 1.88 times higher Il-10 levels than the HCD groups and the HFD group showed 1.55 and 1.66 times higher Il-10 levels than the WT and ND groups respectively. The significant difference was observed only between HFD and HCD groups comparison (P<0.01) (Fig 5D).

## 3.3. Combination of IBD-affecting factors exacerbate inflammation and loosen the intestinal barrier

Based on the histopathological changes between each group, NF-κB and Claudin-2 were analyzed. First, the activated form of NF-κB, a ubiquitous transcription factor activated in

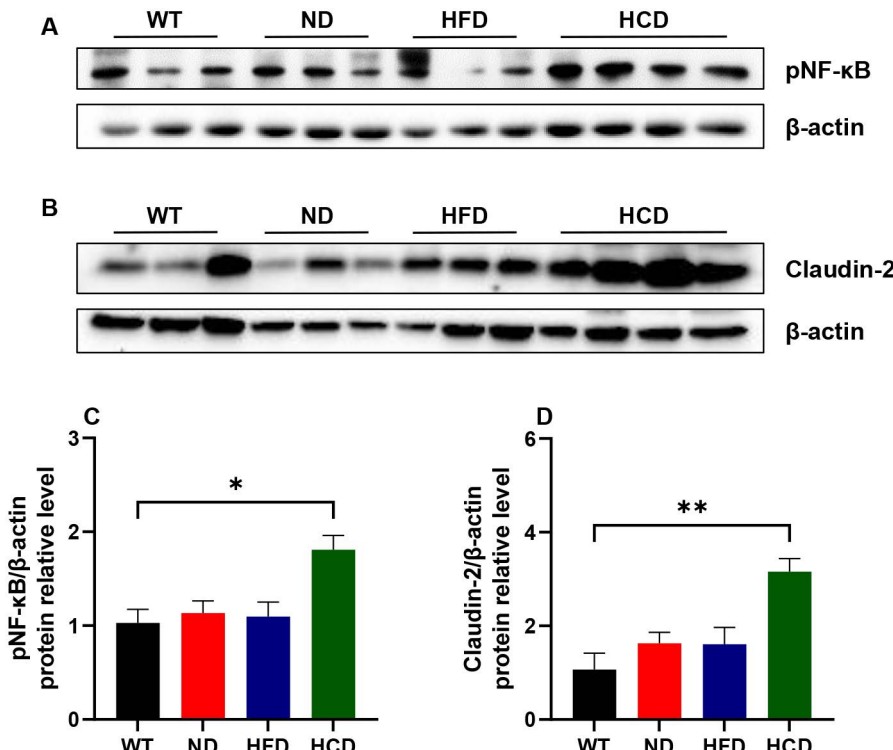

**Fig 6. The expression of phospho-NF-κB p65 and Claudin-2.** Representative images of phospho-NF-κB p65 (A) and Claudin-2 (B) expression in the colon by western blot analysis. The relative densities of phospho-NF-κB p65 (C) and Claudin-2 (D) were calculated relative to β-actin expression. The data for each group is represented as the mean ± SEM. (n=6–7) *P < 0.05; **P < 0.01. (Kruskal-Wallis test).

inflammatory status, was increased when inflammation worsened. The HCD group showed 1.76, 1.59, and 1.65 times higher levels than the WT, ND, and HFD groups respectively, but significant difference was observed only in WT and HCD group comparison (P<0.05) ([Fig 6C]). Second, Claudin-2, the tight junction marker that indicates the leakage in the intestinal barrier, was also increased in the same trend as NF-κB. The HCD group showed 2.95, 1.94, and 1.96 times higher levels than the WT, ND, and HFD groups respectively, but significant difference was observed only in WT and HCD group comparison (P<0.01) ([Fig 6D]).

### 3.4. Changes in the gut microbiota due to factors affecting IBD

As a combination of genetic factors, Westernized diet, and other environmental factors imitate IBD in humans, we investigated whether these combinations also affect the gut microbiota. Based on a previous study in which the outer mucus layer of the colon had more abundant microbiota than the inner mucus layer [30], the colon was opened longitudinally and sampled aseptically. Each outer mucus sample was sequenced and analyzed for the 16s rRNA (V3-V4) hypervariable region.

Alpha diversity, including the Shannon and Faith's phylogenetic diversity (Faith-PD) indexes, was performed to identify the diversity in each group that varied with the IBD factors. The ND, HFD, and HCD groups showed lower Shannon index than the WT group. The HCD group showed significantly lower values than those in the other groups, and the HFD and ND groups showed significantly lower values than those in the WT group (P<0.01) ([Fig 7A]). The WT and ND groups showed similar values in the Faith-PD index, which considers

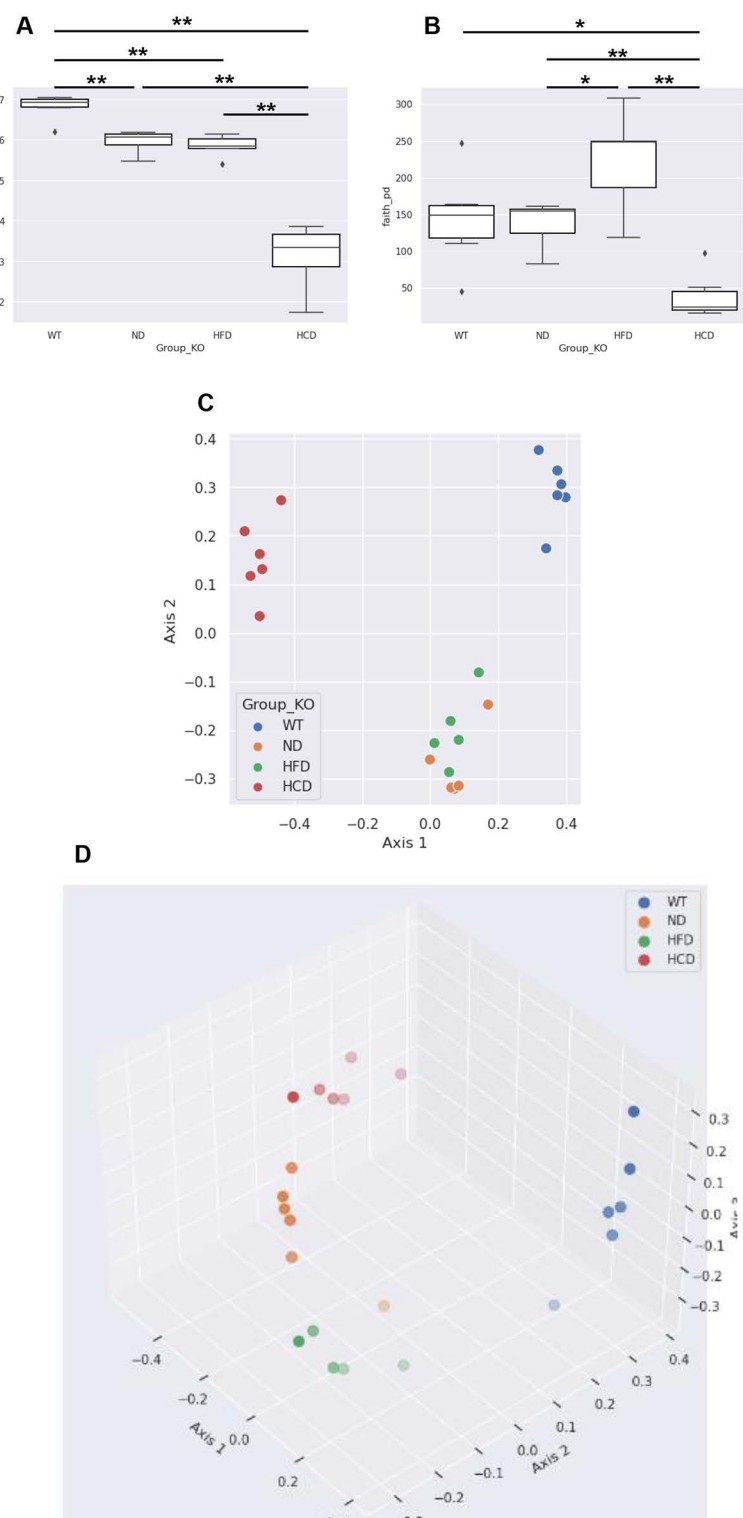

**Fig 7. Characteristics of the bacterial community between groups.** Comparison of the colonic mucus bacterial community between groups. (**A**) Shannon index of each group; (**B**) Faith-PD index of each group; (**C**) Bray–Curtis distance of each group, 2D; (**D**) Bray–Curtis distance of each group, 3D. Data of quantitative analyses are represented as the mean ± SEM. (n = 6–7) *P < 0.05; **P < 0.01. (Alpha diversity: Kruskal–Wallis test; Beta diversity: PERMANOVA).

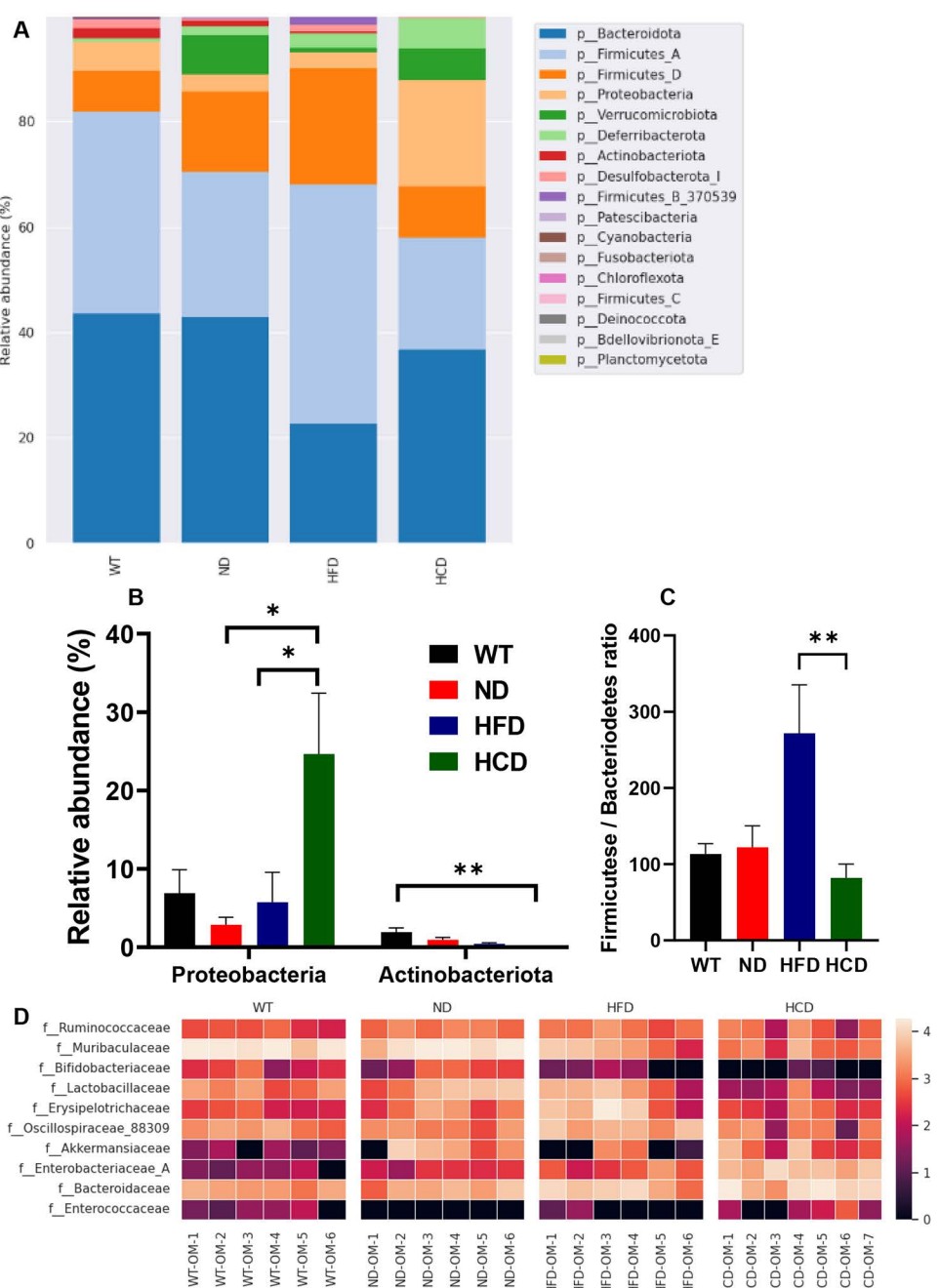

**Fig 8. Relative abundance analysis between groups.** Relative abundance analysis of the colonic mucus bacterial community between groups.(A) Bar plot of the relative abundance at the phylum level of each group; (B) Bar plot of the relative abundance of Proteobacteria and Actinobacteriota; (C) Firmicutes/Bacteroidetes (F/B) ratio between 4 different groups; (D) Heatmap of abundance variation of 10 selected common bacterial taxa in colonic mucus of each group. The relative abundance of selected taxa is based on LEfSe, in which the LDA score is >3.0. Black represents less abundance, red represents intermediate abundance, and white represents most abundance. Data from quantitative analyses are represented as the mean ± SEM (n=6–7) *P < 0.05; **P < 0.01; ***P < 0.001 (Kruskal-Wallis test).

phylogenetic distance. However, the HFD group showed higher values, and the HCD group showed lower values than those in the WT and ND groups. The HCD group showed significantly lower values than those in the WT, ND, and HFD groups (P<0.05, P<0.01, and P<0.01, respectively). However, the HFD group showed significantly higher values than the ND group (P<0.05) (Fig 7B). Based on Bray–Curtis dissimilarity, beta diversity was analyzed by drawing PCoA plots in 2D (Fig 7C) and the 3D (Fig 7D). Regarding Bray–Curtis dissimilarity, significant differences in distance were observed between all groups (P<0.001) (Fig 7C and 7D).

In the bar plot based on relative abundance at the phylum level (L2), the main phyla were Bacteroidetes and Firmicutes, followed by Proteobacteria, Deferribacterota, Verrucomicrobiota, Desulfobacterota, Actinobacteriota, Patescibacteria, Cyanobacteria, Fusobacteriota, Chloroflexota, Deinococcus, Bdellovibrionota, and Planctomycetota (Fig 8A). Proteobacteria, which is known to exhibit increased abundance in CD patients, increased in the HCD group. The HCD group showed 3.58, 8.58, and 4.29 times higher levels than the WT, ND, and HFD groups, respectively. Significant differences were observed in the ND and HFD groups compared to the HCD group (P<0.05). Actinobacteriota which reduced in CD and UC patients, decreased in the HCD group. The HCD group showed 48.57, 23.78, and 11.0 times lower levels than the WT, ND, and HFD groups respectively, The significant differences were observed only in WT group compared to the HCD group (P<0.05) (Fig 8B). Based on previous studies on the correlation between IBD and changes in the ratio of Firmicutes to Bacteroidetes [31, 32]. The ratio of the Firmicutese/Bacteriodetes (F/B ratio) were present in the WT (113.25), ND (122.27), HFD (271.80), and HCD groups (81.93). F/B ratio significantly increased in the HFD group than HCD group (P<0.01) (Fig 8C). Differentially abundant operational taxonomic units (OTU) between each group were identified using linear discriminant analysis Effect Size (LEfSe). Among the OTUs with a linear discriminant analysis (LDA) score of ≥3.0, a heat map was generated by selecting a family associated with IBD. In total, 10 families were selected: Ruminococcaceae, Muribaculaceae, Bifidobacteriaceae, Lactobacillaceae, Erysipelotrichaceae, Oscillospiraceae, and Akkermansiaceae, which are known to be decreased in IBD conditions, and Enterobacteriaceae, Bacteroidaceae, and Enterococcaceae, which are known to be increased in IBD conditions. After analyzing these families for each group, a heatmap was drawn for the six observations. First, the relative abundance of Muribaculaceae and Bifidobacteriaceae gradually decreased as factors known to affect IBD were applied. Second, the relative abundance of Enterobacteriaceae and Bacteroidaceae increased as factors known to affect IBD were applied. Third, in all groups except the WT group, the relative abundances of Ruminococcaceae and Lactobacillaceae decreased, which are factors known to affect IBD. In the WT group, the relative abundances of Ruminococcaceae and Lactobacillaceae decreased regardless of factors affecting IBD. Fourth, in all groups except the WT group, the relative abundance of Enterococcaceae increased as factors known to affect IBD were applied. In the WT group, the relative abundance of Enterococcaceae was higher than in the ND and HFD groups, but lower than in the HCD group, regardless of factors affecting IBD. Fifth, the relative abundances of Erysipelotrichaceae and Oscillospiraceae were lower in the HCD group than in the other groups but higher in the HFD group than in the other groups. Sixth, the relative abundance of Akkermansiaceae was lower in the HCD group than in the ND group. In contrast to the distribution in the ND and HCD groups, it was rarely present in the WT group and was only a minority in the HFD group compared with the ND and HCD groups (Fig 8D).

## 4. Discussion

Since UC, and CD first reported in the early 1900s [33, 34], IBD has emerged as a severe threat to public health in modern society. Fundamental research and updates on therapeutic guidelines are active to reduce its global public health burden [9, 35]. A complex interaction of

multiple factors influences the development of human IBD. However, current animal models of IBD often focus on specific factors, which limits their ability to fully understand the pathogenesis of IBD and their application in therapeutic research. For instance, genetic models of IBD in animals typically study the direct effect of a single gene, while in humans, disease risk is rarely associated with the complete loss of function of a single gene or protein [36, 37]. The limitation of chemically induced IBD models is inducing self-limiting inflammation rather than a chronic phase of inflammation, which mimics human pathogenesis [38]. Another example, diet-induced IBD models, have limitations in that models couldn't cause IBD alone but must be applied together with other factors [12]. Novel animal models mimicking the complex pathogenesis of IBD in humans are required to overcome these limitations. However, no models that complexly mimic the etiology of human IBD have been developed yet. This study aimed to mimic the complex pathogenicity of human IBD combining various IBD factors in mice. Our results provide evidence that this novel complex IBD model reproduces human IBD and overcomes the limitations of the previous IBD models.

The severity of inflammation increased with the combination of IBD factors. As more factors were added to the mouse model, inflammation worsened and the colon shortened. Although Il2rg-deficient mice did not show pathological findings alone, mild inflammation in the HFD group was localized to the mucus layer, and the HCD group showed transmural inflammation and epithelial ulceration. Transmural inflammation and ulceration are the representative characteristics consistent with CD, the chronic IBD of the human [39]. These findings suggest that the limitation of the chemical-induced models, self-limiting inflammation, has been overcome due to a combination of IBD factors. Also, IBD is exacerbated through the combined application of IBD factors rather than focusing on the genetically modified model and its effects on the gene, suggesting that the limitations of the existing genetic model were overcome. Moreover, a key indicator of neutrophil infiltration in IBD, MPO was dramatically increased in the colon of the HCD group. This result is consistent with the previous study, which reported markedly increasing MPO levels in active IBD patients [40].

Cytokines, transcription factor, and tight junction protein associated with inflammation changed depending on the combination of IBD factors. Similar to the histopathological finding, the pro-inflammatory cytokines IFN-γ, Il-1β, Il-6, TNF-α, and CXCL-10 increased with the complex application of the IBD factor. TNF-α, a cytokine crucial in IBD both CD and UC, promotes the production of inflammatory cytokines and induces the death of intestinal epithelial cells [41]. Like TNF-α, the expression of Il-1β, and CXCL10 also increases in IBD conditions [42, 43]. IFN-γ, which plays a major role in CD pathogenesis [44], did not show significant differences in tissue mRNA level but in serum protein level. These conflicting results between mRNA and protein are consistent with previous studies suggesting that due to a synthesis delay between mRNA and protein [45]. Anti-inflammatory cytokines secreted from T cells, Il-10, appeared at lower levels in the HCD group and higher levels in the HFD group. This difference in IL-10 levels between the HFD and HCD groups is consistent with previous studies in which up-regulated IL-10 was associated with the resolution of self-limiting inflammation and the prevention of chronic inflammation The results from the HFD and HCD groups were consistent with previous studies showing that the Il-10 level was associated with worsening or resolving the disease [46]. Based on these results, it can be inferred that the HFD group with the application of chemical stress and dysbiosis, can cause the disease to progress from a self-limiting nature to a chronic form. Notably, the level of Il-6 remained low depending on the common gamma chain deletion. A direct correlation between Il-6 and the common gamma chain has not been revealed; however, previous studies show that Il-6 cooperates with common gamma chain-related cytokines, such as Il-2, Il-7, and Il-15, for T-cell activation [47]. Based on this previous study, it was deduced that Il-6 downregulation

results from the absence of the common gamma chain. NF-κB, known to be upregulated in CD and UC, is a transcription factor involved in inflammatory and immune responses. After NF-κB activation by the pro-inflammatory mediator including IFN-γ, it activates transcription of various genes such as Il-1, Il-6, TNF-α, and CXCL-10 [48]. Another important marker, claudin-2 is a marker of tight junctions that indicates the permeability of the intestinal barrier. In both CD and UC, Claudin-2 is upregulated and exacerbates inflammation by increasing intestinal bacterial permeation [49, 50]. Upregulation of NF-κB and claudin-2 confirmed in the HCD group. Despite the dramatic trends observed in the HCD group, some groups did not show significant differences in mRNA and protein analyses. This may be attributed to the use of nonparametric tests due to the limited sample size. Nevertheless, the observed patterns of pro- and anti-inflammatory cytokines, transcription factors, and tight junction proteins indicate that the combination of complex IBD factors successfully mimics human IBD pathogenesis.

Alterations in the gut microbiota appear to depend on a combination of IBD factors. Regarding alpha diversity, genetic factors did not cause phylogenetic differences but did cause differences in species diversity. The HFD group did not show a significant difference in species diversity compared with the ND group but showed significantly increased biodiversity in phylogenetic differences. This result is consistent with previous studies showing that ingesting HFD changes the gut microbiota of mice, regardless of whether the mice develop obesity [51]. However, despite being fed the same high-fat diet as the HFD group, the HCD group showed deprived biodiversity in both species and phylogenetic diversities compared with all other groups. Based on previous studies showing that dysbiosis occurred after administering DSS to healthy mice [52], we inferred that the combined administration of DSS and *C. rodentium* might lower alpha diversity. In addition, the beta diversity of the WT, ND, HFD, and HCD groups differed significantly. These results may be attributable to various IBD factors. In the relative abundance analysis, the HCD group exhibited relatively higher levels of Proteobacteria and lower levels of Actinobacteriota compared to the other groups. These results align with previous studies indicating that Proteobacteria is more abundant in CD patients, while Actinobacteriota is less abundant in both CD and UC patients [53]. In the relative abundance analysis, dysbiosis was identified through an imbalance between Firmicutes and Bacteroidetes. In our results, the HFD group showed high F/B ratio, whereas the HCD group showed low F/B ratio. These results correspond with those of previous studies, which stated that high F/B ratio indicated obesity condition and low F/B ratio indicated IBD condition [32]. Based on the relative abundance analysis results, Ruminococcaceae, Muribaculaceae, Bifidobacteriaceae, Lactobacillaceae, Erysipelotrichaceae, Oscillospiraceae, and Akkermansiaceae were the dominant families under healthy conditions, and Enterobacteriaceae, Bacteroidaceae, and Enterococcaceae were the dominant families under IBD conditions. This result is consistent with previous studies showing that Ruminococcaceae, Bifidobacteriaceae, Erysipelotrichaceae, Oscillospiraceae, and Akkermansiaceae were more abundant in healthy individuals, and Enterobacteriaceae, Bacteroidaceae, and Enterococcaceae were more abundant in patients with CD and UC [54–61]. In the HFD group, the levels of Akkermansiaceae and Erysipelotrichaceae were markedly lower and higher, respectively. Erysipelotrichaceae decreased in severe IBD conditions and increased with HFD application. These results are consistent with studies that Erysipelotrichaceae decreased in UC patients [53] and increased in obese conditions compared to healthy controls [62]. Additionally, the result for Akkermansiaceae is consistent with previous findings showing that the number of this family decreased in obese conditions [63]. Muribaculaceae, which are more abundant in mice than in other species, are dramatically reduced in mouse IBD models [64, 65]. Humans are the species known to have the highest levels of Muribaculaceae after mice, but research on patients with IBD is still ongoing. The evaluation of Lactobacillaceae is conflicting. Contrary to

our findings that Lactobacillaceae are depleted when IBD factors are applied, Lactobacillaceae are reported to increase in patients with CD and UC [54, 56–59]. However, other studies that correspond to our results have shown that Lactobacillaceae are beneficial bacteria and can be a potential treatment for IBD [66, 67]. Rumincoccaceae, Lactobacillaceae, and Akkermansiaceae were lower in the WT group than in the Il2rg-deficient group, and Enterococcaceae were higher in the WT group than in the Il2rg-deficient group. These results are contrary to our expectations. These results, despite sufficient acclimation, are thought to be due to genetic differences [68]. Little research has been conducted on the correlation between common gamma chain deficiency and the levels of Ruminococcaceae, Lactobacillaceae, Akkermansiaceae, and Enterococcaceae. Therefore, further studies are required to determine these correlations.

This study's data demonstrated that novel complex IBD models that mimic human IBD pathogenesis developed successfully. Despite developing several animal models for IBD therapeutic and pathogenesis research, current animal models have limitations in imitating the complex IBD pathogenesis. Our study revealed that animal models combining genetic traits and environmental factors, such as a Westernized diet, bacteria, and chemical stress, successfully mimic human IBD, especially CD, regarding chronic colitis and microbiota dysbiosis more closely than current models. In particular, this study demonstrated that the limitations of the existing IBD model were overcome by improving the current IBD model into a novel complex IBD model. Our novel complex IBD models could be valuable tools for IBD research, including the development of new therapeutic strategies and the elucidation of complex IBD pathogenesis.

## Supporting information

**S1 Raw image. Uncropped images of blot presented in Fig 6A & 6B.**
(PDF)

## Author contributions

**Conceptualization:** Sun-Min Seo, Yang-Kyu Choi.

**Data curation:** Sun-Min Seo.

**Formal analysis:** Sun-Min Seo.

**Funding acquisition:** Yang-Kyu Choi.

**Investigation:** Sun-Min Seo, Na-Won Kim, Eun-Seon Yoo, Ji-Hun Lee, Ah-Reum Kang, Han-Bi Jeong, Won-Yong Shim, Dong-Hyun Kim, Young-Jun Park, Kieun Bae.

**Methodology:** Sun-Min Seo, Yang-Kyu Choi.

**Project administration:** Yang-Kyu Choi.

**Supervision:** Kyong-Ah Yoon, Yang-Kyu Choi.

**Writing – original draft:** Sun-Min Seo.

**Writing – review & editing:** Yang-Kyu Choi.

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
