## [Decision Letter · Decision Letter 0]

8 Mar 2024

PONE-D-24-05316Development of novel complex inflammatory bowel disease mouse models: reproducing various human inflammatory bowel disease etiologies in micePLOS ONE

Dear Dr. Choi,

Thank you for submitting your manuscript to PLOS ONE. After careful consideration, we feel that it has merit but does not fully meet PLOS ONE’s publication criteria as it currently stands. Therefore, we invite you to submit a revised version of the manuscript that addresses the points raised during the review process.

The major concerns remain about the lack of appropriate statistical analysis, failure to outline the limitations in previous models and how those have been overcome in the authors models.  In addition, poor presentation of results, figures as well as grammatical errors further dampen the enthusiasm.  Authors need to address the concerns of both the reviewers in a point by point manner.

We look forward to receiving your revised manuscript.

Kind regards,

Pradeep Dudeja

Academic Editor

PLOS ONE

Journal Requirements:

"This work was supported by the Bio & Medical Technology Development Program of the National Research Foundation (NRF) funded by the Korea government (MSIT) (Nos. 2020R1A2C2005898 and 2021M3H9A1097269) and the Konkuk University Researcher Fund in 2023."

Reviewers' comments:

Reviewer's Responses to Questions

**Comments to the Author**

1. Is the manuscript technically sound, and do the data support the conclusions?

Reviewer #1: Partly

Reviewer #2: No

2. Has the statistical analysis been performed appropriately and rigorously? 

Reviewer #1: Yes

Reviewer #2: No

3. Have the authors made all data underlying the findings in their manuscript fully available?

Reviewer #1: Yes

Reviewer #2: No

4. Is the manuscript presented in an intelligible fashion and written in standard English?

Reviewer #1: No

Reviewer #2: Yes

5. Review Comments to the Author

Reviewer #1: The goal of this paper is to develop complex models that mimic IBD in humans more closely than currently available models. This is a descriptive paper focusing on some observations and does not delve into possible mechanisms.

The authors are requested to address the following comments and provide clarifications.

1. How many new models have been developed? Two? HFD and HCD? State this in the title.

2. Throughout the text authors make generalized statements without providing specifics. This should be avoided. For example, they consistently state that present models have limitations and while they provide references, they do not elaborate on what are the specific limitations. If the presented models are better, they need to specifically state how it overcomes the limitations of other models and give pertinent details of the other models.

3. Line 153-154: What happens if WT animals are fed a HFD? Do they develop any signs of inflammation as the Il2rg deficient mice?

4. Line 264: Unclear what is meant by 6 types – the six observations?

5. Line 310-312: Are there no other models that show these characteristics of inflammation? What makes the present model unique?

6. Line 315; When the authors state “gradually increased …” did they perform a time course to measure the increase in cytokine expression?

7. Line 317-318: If interferon gamma did not increase as anticipated, how do the authors reconcile this as being a good model for IBD.

8. Line 324-325: How do the authors reconcile the increase in IL-10 in the HFD group with the latter being a model for IBD (MC). IF this is information in Ref 51, it should be explained better.

9. Line 331-332: Do the authors conclude that HCD mimics CD and HFD mimics MC based strictly on PCR results of cytokine expression?

10. Line 347-350: The statements made on high vs low levels are difficult to follow with the date shown in Figure 4. For example while the figure shows low Firmicutes in HCD, it does not necessarily show high Bacteroidetes (the levels in HCD and ND are comparable.

11. Line 359-360: If Erysipelotrichaceae is high in the HFD group, then how does it reconcile with the previous sentence which states that these species are abundant in healthy individuals. Please clarify.

12. Line 375: A stronger argument with more convincing comparisons to other models and data are needed to support this conclusion.

13. Line 381-382, the authors state the HCD model is good for the study of human CD and UC and yet the discussion on the data does not provide evidence on the connection to UC.

Figures: The practice of embedding the figure legends in the text is not common and therefore a bit difficult to follow. The maximal details should be provided in the methods and those in the legend and figure should reflect the highlights. For example, the methods do not state how DSS was administered, but it is only stated in the figure 1.

All figure legends should state the “n” values even if they are the same for each experiment.

Figure 1: The methods should clearly reflect what was done. Lines 62-90 were confusing to this reviewer. It could be interpreted as 56+17 = 73 days (which seems to be the case based on the figure) or as 56days on ND +56days on diet +17 days = 129 days. As written in this and previous page it could be either. Is “0” day, the day of birth or are they 8 weeks old at that stage. If the former, when were the animals weaned from their dams?

Line 153-154 states that figure 1 shows “genetic factors were imitated….” This reviewer could not find that information in the figure. Please explain.

Figure 4: As done for figure 2 and 3, please indicate the cytokine being measured in the y-axes; it is difficult to follow if one has to refer back to the text.

Figure 6: Please label Y-axes in panels B and C, as stated for Figure 4.

Editorial: The paper should be carefully reviewed for editorial and grammatical inconsistencies. A few examples are provided and this list is not complete

line 43, should read “… Clostridium difficile infections…”;

line 65, the expansion of HCD should be provided when it is first used here;

Line 94-97: the sentence is repeated;

line 103: What is meant by “….performed independently on three individuals” – samples from three animals were analyzed or does it mean that the histopathological scoring and analyses (double blind) were conducted by three separate researchers?

Line 303: Do the authors mean altered bacterial environment (including composition) when they state “bacteria”?

Reviewer #2: In this study, the authors have developed the mouse model of IBD using various human IBD etiologies. However, many major concerns need to be addressed.

Major points:

• A change in the myeloperoxidase enzyme activity must support the change in the colon length.

• A complete picture of the Swiss roll must be provided along with the magnified inserts. The scale bar must be included in the picture.

• What does the bar diagram Fig 2B indicating the histopathological scoring indicate?

• The Fig2C bar diagram showing HFD and HCD standard deviation (SD) is very high. With this high SD, having a statistical significance in what is depicted is unrealistic. Therefore, statistics analysis must be revisited.

• To evaluate the reliability of the newly established IBD model, several parameters of key anti-inflammatory activity must be assessed. These include LI-6, IL-1B, TNF-alpha, and CXCL-10 protein concentration changes, preferably through Western Blot analysis.

• One of the key proinflammatory transcription factor that is activated is NF-κB; therefore, authors must investigate its expression of it.

• One of the key events that is affected is the breach of the tight junction. Therefore, colon permeability along with tight junction protein expression must be investigated before it can be acceptable as a model that mimics IBD with complex etiology.

6. PLOS authors have the option to publish the peer review history of their article (what does this mean? ). If published, this will include your full peer review and any attached files.

**Do you want your identity to be public for this peer review?** For information about this choice, including consent withdrawal, please see our Privacy Policy .

Reviewer #1: No

Reviewer #2: No

---

## [Author Response · Author response to Decision Letter 0]

9 Jun 2024

Pradeep Dudeja

Academic Editor

PLOS ONE

Dear Dr. Dudeja 

Subject: Development of a novel complex inflammatory bowel disease mouse model: reproducing human inflammatory bowel disease etiologies in mice [PONE-D-24-05316R1]

Thank you for inviting us to submit a revised draft of our manuscript entitled, " Development of a novel complex inflammatory bowel disease mouse model: reproducing various human inflammatory bowel disease etiologies in mice" to PLOS ONE. We also appreciate the time and effort you and each of the reviewers have dedicated to providing insightful feedback on ways to strengthen our paper. Thus, it is with great pleasure that we resubmit our article for further consideration. We have incorporated changes that reflect the detailed suggestions you have graciously provided. We also hope that our edits and the responses we provide below satisfactorily address all the issues and concerns you and the reviewers have noted. Also, we declare that the funders had no role in study design, data collection and analysis, decision to publish, or preparation of the manuscript.

To facilitate your review of our revisions, the following is a point-by-point response to the questions and comments delivered in your letter dated 05 June 2024.

Proposals from Academic Editor

Thank you for submitting your manuscript to PLOS ONE. After careful consideration, we feel that it has merit but does not fully meet PLOS ONE’s publication criteria as it currently stands. Therefore, we invite you to submit a revised version of the manuscript that addresses the points raised during the review process.

The major concerns remain about the lack of appropriate statistical analysis, failure to outline the limitations in previous models and how those have been overcome in the authors models. In addition, poor presentation of results, figures as well as grammatical errors further dampen the enthusiasm. Authors need to address the concerns of both the reviewers in a point by point manner.

Answer: We revised our manuscript based on the PLOS ONE style template.

"This work was supported by the Bio & Medical Technology Development Program of the National Research Foundation (NRF) funded by the Korea government (MSIT) (Nos. 2020R1A2C2005898 and 2021M3H9A1097269) and the Konkuk University Researcher Fund in 2023." Please state what role the funders took in the study.

If the funders had no role, please state: "The funders had no role in study design, data collection and analysis, decision to publish, or preparation of the manuscript." If this statement is not correct you must amend it as needed. 

Answer: We added “role of funder statement” in our cover letter.

Comments from Reviewer #1:

The goal of this paper is to develop complex models that mimic IBD in humans more closely than currently available models. This is a descriptive paper focusing on some observations and does not delve into possible mechanisms. The authors are requested to address the following comments and provide clarifications.

1. How many new models have been developed? Two? HFD and HCD? State this in the title.

Answer: Additional experiments requested by Reviewer 2 confirmed that the HFD group was not suitable for the new IBD model. For this reason, we revised the entire manuscript and presented only HCD as a new model. The title has also been modified.

2. Throughout the text authors make generalized statements without providing specifics. This should be avoided. For example, they consistently state that present models have limitations and while they provide references, they do not elaborate on what are the specific limitations. If the presented models are better, they need to specifically state how it overcomes the limitations of other models and give pertinent details of the other models.

Answer: Thank you for your assessment. As the reviewer pointed out, the pre-revision manuscript concluded with a logical leap, without explaining the limitations of the previous IBD model and how those limitations were overcome. Based on these assessment, we revised the manuscript to determine how our results mimic human IBD, what the limitations of previous IBD models were, and how to overcome these limitations. The current IBD models are mainly chemically induced models or models using genetically modified animals. The biggest limitation of these models is that they are difficult to induce chronic features in human IBD and, unlike genetically modified models, have complex pathogenesis in humans. The advantage of our model is that it not only causes chronic inflammation but also overcomes the limitations of each IBD model by mixing various IBD factors rather than focusing on each IBD factor. 

3. Line 153-154: What happens if WT animals are fed a HFD? Do they develop any signs of inflammation as the Il2rg deficient mice?

Answer: Based on our previous pilot study results, WT did not show signs of inflammation following high-fat diet feeding.

4. Line 264: Unclear what is meant by 6 types – the six observations?

Answer: Thank you for your suggestion. As the reviewer asked, “6 observation” is more appropriate than “6 types”, so we revised the corresponding word.

5. Line 310-312: Are there no other models that show these characteristics of inflammation? What makes the present model unique?

Answer: Characteristics of inflammation, such as ulcer and transmural inflammation seen in our model are also present in the TNBS-induced IBD model. However, TNBS induced colitis does not recapitulate the disease in terms of etiopathogenesis. For this reason our model have a unique advantage.

6. Line 315; When the authors state “gradually increased …” did they perform a time course to measure the increase in cytokine expression?

Answer: We meant “gradually increased” as inflammation worsened by the IBD factors added one by one. However, because there may be misunderstanding in interpretation, “gradually” was deleted from the sentence. Additionally, the same expressions were also deleted from the manuscript.but this was corrected because there may have been misunderstandings in the interpretation. 

7. Line 317-318: If interferon gamma did not increase as anticipated, how do the authors reconcile this as being a good model for IBD.

Answer: Thank you for providing these insights. We performed ELISA on serum levels of cytokines and confirmed that interferon gamma showed a significant difference in the HCD group, unlike mRNA level. These conflicting results between mRNA and protein are consistent with previous studies suggesting that due to a synthesis delay between mRNA and protein. These result available in revised Fig 5. additional results in (p. 13, line 254-255), and discussion in (p. 19, line 391-394 with additional reference [46]) (Added and revised sentences marked with red color).

8. Line 324-325: How do the authors reconcile the increase in IL-10 in the HFD group with the latter being a model for IBD (MC). IF this is information in Ref 51, it should be explained better.

Answer: Additional experiments requested by Reviewer 2 confirmed that the HFD group was not suitable for the new IBD model. We concluded that IL-10 upregulation in the HFD group indicates the resolution of self-limiting inflammation, not becoming chronic form. This information is written in (p .19 line 395-400).

9. Line 331-332: Do the authors conclude that HCD mimics CD and HFD mimics MC based strictly on PCR results of cytokine expression?

Answer: Additional experiments requested by Reviewer 2 confirmed that the HFD group was not suitable for the new IBD model. However, unlike HFD, combining protein level and mRNA level results of cytokine confirms HCD mimics CD.

10. Line 347-350: The statements made on high vs low levels are difficult to follow with the date shown in Figure 4. For example while the figure shows low Firmicutes in HCD, it does not necessarily show high Bacteroidetes (the levels in HCD and ND are comparable.

Answer: Thank you for your insight. To reflect the difficulty in understanding the plots, the figure and results were changed to F/B ratio to confirm the relative abundance of Firmicutese and Bacteriodetes more objectively. (p. 16 line 316-320, marked with red color)

11. Line 359-360: If Erysipelotrichaceae is high in the HFD group, then how does it reconcile with the previous sentence which states that these species are abundant in healthy individuals. Please clarify.

Answer: We acknowledge that there was a lack of explanation for the sentence. The previous study shows that Erysipelotrichaceae is low in UC patients and high in obese conditions compared to Healthy control. The manuscript has been revised to include the relevant content. (p. 21 line 436-439, marked with red color)

12. Line 375: A stronger argument with more convincing comparisons to other models and data are needed to support this conclusion.

Answer: Opinions of the relative abundance of Rumincoccaceae, Lactobacillaceae, Akkermansiaceae, and Enterococcaceae between healthy control and IBD patients are controversial. However, when compared with previous research results on the microbiota of human IBD patients, this turned out to be different from our expectations. These changes in outcome are thought to be due to deletion of IL2rg, also known as the common gamma chain. The previous study that genetic background has a stronger impact on the composition of the microbiota than maternal inoculation or exogenous microbiota, which corresponded to our results. We revised the sentence and add a reference. (p. 21, line 448-449, additional reference [69], marked with red color.)

13. Line 381-382, the authors state the HCD model is good for the study of human CD and UC and yet the discussion on the data does not provide evidence on the connection to UC.

Answer: Thank you for your assessment. Based on the additional experimental results, the manuscript was revised as a whole, and the significant differences in TNF-α and NF-κB were indicated as evidence of the connection to UC.

14. Figures: The practice of embedding the figure legends in the text is not common and therefore a bit difficult to follow. The maximal details should be provided in the methods and those in the legend and figure should reflect the highlights. For example, the methods do not state how DSS was administered, but it is only stated in the figure 1.

Answer: Thank you for your suggestion. As your advice we checked the missing parts in the method and added the necessary information. The information about the administration of DSS for Fig 1, provided as an example is written in (p. 4 line 72-73, marked with red color) 

15. All figure legends should state the “n” values even if they are the same for each experiment.

Answer: As your advice, we indicated “n” values in all figures.

16. Figure 1: The methods should clearly reflect what was done. Lines 62-90 were confusing to this reviewer. It could be interpreted as 56+17 = 73 days (which seems to be the case based on the figure) or as 56days on ND +56days on diet +17 days = 129 days. As written in this and previous page it could be either. Is “0” day, the day of birth or are they 8 weeks old at that stage. If the former, when were the animals weaned from their dams?

Answer: We agree that our expression was ambiguous. Our experimental design was 56+17=73 days. So we revised sentences that might cause confusion (p. 4 line 67-70, marked with red color). Additionally, day "0" is the start date of the experiment, when the mouse is 8 weeks old.전자인 

17. Line 153-154 states that figure 1 shows “genetic factors were imitated….” This reviewer could not find that information in the figure. Please explain.

Answer: Thank you for pointing out the error. The relevant information appears in the previous study, so we revised the sentence and added a reference. (p. 10 line 184-185, marked with red color)

18. Figure 4: As done for figure 2 and 3, please indicate the cytokine being measured in the y-axes; it is difficult to follow if one has to refer back to the text.

Answer: As your advice, we indicated name of the target cytokine on the Y axis.

19. Figure 6: Please label Y-axes in panels B and C, as stated for Figure 4.

Answer: As your advice, we indicated label of the target cytokine on the Y axis. Fig 6. was changed to Fig 8. Because of the addition of other figures.

Comments from Editor:

The paper should be carefully reviewed for editorial and grammatical inconsistencies. A few examples are provided and this list is not complete

1. line 43, should read “… Clostridium difficile infections…”;

Answer: We revised sentence (p. 3, line 45) and marked it with red color.

2. line 65, the expansion of HCD should be provided when it is first used here;

Answer: In addition to the HCD you advised, we also revised full name of WT, ND, and HFD. The revised sentences marked with red color (p. 4 line 62-65, p. 5 84-88)

3. Line 94-97: the sentence is repeated;

Answer: We checked the repeated sentences and deleted them.

4. line 103: What is meant by “….performed independently on three individuals” – samples from three animals were analyzed or does it mean that the histopathological scoring and analyses (double blind) were conducted by three separate researchers?

Answer: Thank you for your suggestion. We describe this sentence to mean that scoring and analyses were conducted by three separate researchers. We recognized that sentence is ambiguous, so we modified it and marked with red color. (p. 6, line 104) 

5. Line 303: Do the authors mean altered bacterial environment (including composition) when they state “bacteria”?

Answer: We agree that our expression was ambiguous. As the editor pointed out, "bacteria" means a change in the bacterial environment, also known as dysbiosis. Due to the extensive revision of the Discussion, the relevant content disappeared, but the ambiguous expression in the manuscript was corrected to dysbiosis.

Comments from Reviewer #2:

In this study, the authors have developed the mouse model of IBD using various human IBD etiologies. However, many major concerns need to be addressed.

1. A change in the myeloperoxidase enzyme activity must support the change in the colon length.

Answer: Thank you for your suggestion. As a result of MPO activity assay, it was confirmed that MPO activity increased as the colon length decreased. Those results are available in revised Fig 3. additional results in (p. 11, line 209-214), and discussion in (p. 19, line 383-385) (Added and revised sentences marked with red color).

2. A complete picture of the Swiss roll must be provided along with the magnified inserts. The scale bar must be included in the picture.

Answer: We added entire picture of swiss roll in Fig 3. Also, the scale bar added in both low and high magnification pictures.

3. What does the bar diagram Fig 2B indicating the histopathological scoring indicate?

Answer: The bar diagram of histopathological scoring is the score calculated based on Table 1. in the manuscript.

4. The Fig2C bar diagram showing HFD and HCD standard deviation (SD) is very high. With this high SD, having a statistical significance in what is depicted is unrealistic. Therefore, statistics analysis must be revisited.

Answer: Thank you for providing this insight. The review made us realize that our statistics were inadequate. Because the number of samples per group was not large enough, the statistical analyses were changed to the Kruskal-Wallis test, and SD was also changed to SEM.

5. To evaluate the reliability of the newly established IBD model, several parameters of key anti-inflammatory activity must be assessed. These include IL-6, IL-1B, TNF-alpha, and CX

---

## [Decision Letter · Decision Letter 1]

6 Aug 2024

PONE-D-24-05316R1Development of a novel complex inflammatory bowel disease mouse model: reproducing human inflammatory bowel disease etiologies in micePLOS ONE

Dear Dr. Choi,

Thank you for submitting your manuscript to PLOS ONE. After careful consideration, we feel that it has merit but does not fully meet PLOS ONE’s publication criteria as it currently stands. Therefore, we invite you to submit a revised version of the manuscript that addresses the points raised during the review process. Both the reviewers felt that the manuscript has significantly improved and authors have addressed almost all of the concerns.  However, one of the reviewer raised some minor concerns which related to issues concerning the comparisons of certain groups and clarification of some queries.  We invite the authors to address these minor concerns and submit a re-revised version. Please submit your revised manuscript by Sep 20 2024 11:59PM. If you will need more time than this to complete your revisions, please reply to this message or contact the journal office at plosone@plos.org . Please include the following items when submitting your revised manuscript:A rebuttal letter that responds to each point raised by the academic editor and reviewer(s). You should upload this letter as a separate file labeled 'Response to Reviewers'.A marked-up copy of your manuscript that highlights changes made to the original version. You should upload this as a separate file labeled 'Revised Manuscript with Track Changes'.An unmarked version of your revised paper without tracked changes. You should upload this as a separate file labeled 'Manuscript'.If applicable, we recommend that you deposit your laboratory protocols in protocols.io to enhance the reproducibility of your results. Protocols.io assigns your protocol its own identifier (DOI) so that it can be cited independently in the future. For instructions see: https://journals.plos.org/plosone/s/submission-guidelines#loc-laboratory-protocols . Additionally, PLOS ONE offers an option for publishing peer-reviewed Lab Protocol articles, which describe protocols hosted on protocols.io. Read more information on sharing protocols at https://plos.org/protocols?utm_medium=editorial-email&utm_source=authorletters&utm_campaign=protocols .

We look forward to receiving your revised manuscript.

Kind regards,

Pradeep Dudeja

Academic Editor

PLOS ONE

Journal Requirements:

Reviewers' comments:

Reviewer's Responses to Questions

**Comments to the Author**

1. If the authors have adequately addressed your comments raised in a previous round of review and you feel that this manuscript is now acceptable for publication, you may indicate that here to bypass the “Comments to the Author” section, enter your conflict of interest statement in the “Confidential to Editor” section, and submit your "Accept" recommendation.

Reviewer #1: All comments have been addressed

Reviewer #3: All comments have been addressed

2. Is the manuscript technically sound, and do the data support the conclusions?

Reviewer #1: Yes

Reviewer #3: Yes

3. Has the statistical analysis been performed appropriately and rigorously? 

Reviewer #1: Yes

Reviewer #3: Yes

4. Have the authors made all data underlying the findings in their manuscript fully available?

Reviewer #1: Yes

Reviewer #3: Yes

5. Is the manuscript presented in an intelligible fashion and written in standard English?

Reviewer #1: Yes

Reviewer #3: Yes

6. Review Comments to the Author

Reviewer #1: The authors have done a good job in revising the manuscript and responding to this reviewer’s concerns. Thank you. There are a few questions –

Page 14, Line 265-66: Can you comment on whether the HFD group was statistically different from WT or ND in mRNA or protein for IL-10?

Page 16-17: Line 332-334: The two sentences appear contradictory – does Enterococcaceae increase in the WT group as suggested by the 2nd sentence?

Page 19 line 396-397: As written here it is unclear whether the change in IL-10 in HFD is associated with worsening or resolving the disease. In responses to the previous review (#8), the authors provide a clearer explanation and I recommend that language should be included here instead. “We concluded that IL-10 upregulation in the HFD group indicates the resolution of self-limiting inflammation, not becoming chronic form. This information is written in (p .19 line 395-400).”

Page 32, Line 410: “This result indicates…..” is an incomplete sentence and should be deleted.

Reviewer #3: The authors have addressed most of the concerns raised by the reviewers. However, the authors should consider the following suggestions:

1. Previous studies have shown an increase in the abundance of Proteobacteria (phylum associated with wide variety of pathogens) in CD patients (Ref#62). Figure 8A shows an increase in the abundance of Proteobacteria in HCD group compared to WT, ND and HFD groups. However, the authors failed to elaborate on this aspect in the revised manuscript. Moreover, further explanation on this aspect would be more convincing to establish that HCD model is good for the study of human CD.

2. Any specific reason as to why HCD gp was not compared to WT and ND groups in Figs. 4D and 5D?

3. Please be consistent. Also compare HCD gp with ND gp in Figs. 4B, 4E, 4F, 6C and 6D.

4. Did the authors examine NF�B (p65) protein levels in nuclear fraction of colon lysates? If yes, NF�B (p65) levels should be normalized to a nuclear protein, Lamin B1

7. PLOS authors have the option to publish the peer review history of their article (what does this mean? ). If published, this will include your full peer review and any attached files.

**Do you want your identity to be public for this peer review?** For information about this choice, including consent withdrawal, please see our Privacy Policy .

Reviewer #1: No

Reviewer #3: No

---

## [Author Response · Author response to Decision Letter 1]

12 Aug 2024

Dear Dr. Dudeja 

Subject: Development of a novel complex inflammatory bowel disease mouse model: reproducing human inflammatory bowel disease etiologies in mice [PONE-D-24-05316R1]

Thank you for inviting us to submit a revised draft of our manuscript entitled, "Development of a novel complex inflammatory bowel disease mouse model: reproducing human inflammatory bowel disease etiologies in mice" to PLOS ONE. We appreciate the time and effort you and the reviewers have dedicated to providing insightful feedback to strengthen our paper. We are pleased to resubmit our article for further consideration, having incorporated the detailed suggestions provided. We believe that our edits and the responses below adequately address all the issues and concerns raised by you and the reviewers.

To facilitate your review of our revisions, the following is a point-by-point response to the questions and comments delivered in your letter dated 12 August 2024.

Comments from Reviewer #1:

The authors have done a good job in revising the manuscript and responding to this reviewer’s concerns. Thank you. There are a few questions

1. Page 14, Line 265-66: Can you comment on whether the HFD group was statistically different from WT or ND in mRNA or protein for IL-10?

A: Thank you for providing these insights. Although the WT and ND groups show lower Il-10 levels in both mRNA and protein analyses, no significant differences were observed compared to the HFD group. These results may be attributed to the use of nonparametric tests due to the limited sample size. We have included these additional results (p.13 line 257-260, p.14 line 279-282) and discussion (p.21 line 437-440) in our revised manuscript.

2. Page 16-17: Line 332-334: The two sentences appear contradictory – does Enterococcaceae increase in the WT group as suggested by the 2nd sentence?

A: We agree that our expression was ambiguous. We intended to explain that the relative abundance of Enterococcaceae increased in the ND, HFD, and HCD groups as factors known to influence IBD were applied, whereas the WT group deviated from this trend. The relative abundance of Enterococcaceae in the WT group was higher than in the ND and HFD groups. We have included this additional explanation in our revised manuscript (P.17-18 line 358-361). Additionally, we also revised the sentences concerning Ruminococcaceae and Lactobacillaceae to avoid potential confusion. (P.17 line 356-358)

3. Page 19 line 396-397: As written here it is unclear whether the change in IL-10 in HFD is associated with worsening or resolving the disease. In responses to the previous review (#8), the authors provide a clearer explanation and I recommend that language should be included here instead. “We concluded that IL-10 upregulation in the HFD group indicates the resolution of self-limiting inflammation, not becoming chronic form. This information is written in (p.19 line 395-400).”

A: Thank you for your suggestion. We agree that our expression was confusing, so we revised our manuscript to be more obvious (p. 20 line 423-425).

4. Page 32, Line 410: “This result indicates…..” is an incomplete sentence and should be deleted.

A: Thank you for your advice. As your advice, we deleted the incomplete sentence.

Comments from Reviewer #3:

The authors have addressed most of the concerns raised by the reviewers. However, the authors should consider the following suggestions:

1. Previous studies have shown an increase in the abundance of Proteobacteria (phylum associated with wide variety of pathogens) in CD patients (Ref#62). Figure 8A shows an increase in the abundance of Proteobacteria in HCD group compared to WT, ND and HFD groups. However, the authors failed to elaborate on this aspect in the revised manuscript. Moreover, further explanation on this aspect would be more convincing to establish that HCD model is good for the study of human CD.

A: Thank you for providing these insights. In addition to the increased abundance of Proteobacteria in the HCD group, we also discussed the reduced abundance of Actinobacteriota, which is known to be decreased in CD and UC patients. These results are presented in our revised result (p.17 line 335-341) and discussion (p.21 line 453-457) sections, along with the newly added Fig. 8B.

2. Any specific reason as to why HCD gp was not compared to WT and ND groups in Figs. 4D and 5D?

A: Thank you for your insightful feedback. Although the HCD group showed lower levels of Il-10 in both mRNA and protein analyses, no statistically significant differences were observed compared to the WT and ND groups, so we initially did not include this data. However, following the recommendations of Reviewer 1 and Reviewer 3, we have now added these results to the revised manuscript (p.13 line 257-260, p.14 line 279-282). As mentioned in our response to Reviewer 1's Question 1, the lack of statistical significance may be attributed to the use of non-parametric tests due to the limited sample size.

3. Please be consistent. Also compare HCD gp with ND gp in Figs. 4B, 4E, 4F, 6C and 6D.

A: As in our response to Question 2, no statistically significant differences were observed in Fig. 4B, 4E, 4F, 6C, and 6D, so we initially did not mention these results. However, following the reviewers' recommendations, we have now included a comparison of expression levels and significance between the HCD group and the ND, WT, and HFD groups in Fig. 4, Fig. 5, and Fig. 6 in the revised manuscript. (p. 13 line 243-255, p. 14 line 264-278, p. 15 line 301-306)

4. Did the authors examine NF�B (p65) protein levels in nuclear fraction of colon lysates? If yes, NF�B (p65) levels should be normalized to a nuclear protein, Lamin B1

A: We normalized p65 activation to β-actin because we aimed to assess the overall phosphorylation status of p65 by examining its protein levels in total lysates without distinguishing between the cytoplasm and nucleus. However, while preparing our response to the revision, we realized that our manuscript did not sufficiently mention the phosphorylation of NF-κB, and we had incorrectly described the antibody used. We have now corrected these details in our revised manuscript. (p. 9 line 148, p. 15 line 300, Fig. 6A, Fig 6C and supplemental figure)

Again, thank you for giving us the opportunity to strengthen our manuscript with your valuable comments and queries. We have worked hard to incorporate your feedback and hope that these revisions persuade you to accept our submission.

---

## [Editor Report · Decision Letter 2]

2 Sep 2024

Development of a novel complex inflammatory bowel disease mouse model: reproducing human inflammatory bowel disease etiologies in mice

PONE-D-24-05316R2

Dear Dr. Choi,

We’re pleased to inform you that your manuscript has been judged scientifically suitable for publication and will be formally accepted for publication once it meets all outstanding technical requirements.

An invoice will be generated when your article is formally accepted. Please note, if your institution has a publishing partnership with PLOS and your article meets the relevant criteria, all or part of your publication costs will be covered. Please make sure your user information is up-to-date by logging into Editorial Manager at Editorial Manager®  and clicking the ‘Update My Information' link at the top of the page. If you have any questions relating to publication charges, please contact our Author Billing department directly at authorbilling@plos.org .

If your institution or institutions have a press office, please notify them about your upcoming paper to help maximize its impact. If they’ll be preparing press materials, please inform our press team as soon as possible -- no later than 48 hours after receiving the formal acceptance. Your manuscript will remain under strict press embargo until 2 pm Eastern Time on the date of publication. For more information, please contact onepress@plos.org .

Kind regards,

Pradeep Dudeja

Academic Editor

PLOS ONE
---

## [Editor Report · Acceptance letter]

19 Sep 2024

PONE-D-24-05316R2 

PLOS ONE

Dear Dr. Choi, 

I'm pleased to inform you that your manuscript has been deemed suitable for publication in PLOS ONE. Congratulations! Your manuscript is now being handed over to our production team.

Lastly, if your institution or institutions have a press office, please let them know about your upcoming paper now to help maximize its impact. If they'll be preparing press materials, please inform our press team within the next 48 hours. Your manuscript will remain under strict press embargo until 2 pm Eastern Time on the date of publication. For more information, please contact onepress@plos.org .

If we can help with anything else, please email us at customercare@plos.org .

Kind regards, 

on behalf of

Dr. Pradeep Dudeja 

Academic Editor

PLOS ONE